# Comparative Study of Mortality Rate Prediction Using Data-Driven Recurrent Neural Networks and the Lee–Carter Model

**Yuan Chen \* and Abdul Q. M. Khaliq \***

Department of Mathematical Sciences, Middle Tennessee State University, Murfreesboro, TN 37132-0001, USA
\* Correspondence: yc3y@mtmail.mtsu.edu (Y.C.); abdul.khaliq@mtsu.edu (A.Q.M.K.)

**Abstract:** The Lee–Carter model could be considered as one of the most important mortality prediction models among stochastic models in the field of mortality. With the recent developments of machine learning and deep learning, many studies have applied deep learning approaches to time series mortality rate predictions, but most of them only focus on a comparison between the Long Short-Term Memory and the traditional models. In this study, three different recurrent neural networks, Long Short-Term Memory, Bidirectional Long Short-Term Memory, and Gated Recurrent Unit, are proposed for the task of mortality rate prediction. Different from the standard country level mortality rate comparison, this study compares the three deep learning models and the classic Lee–Carter model on nine divisions' yearly mortality data by gender from 1966 to 2015 in the United States. With the out-of-sample testing, we found that the Gated Recurrent Unit model showed better average MAE and RMSE values than the Lee–Carter model on 72.2% (13/18) and 67.7% (12/18) of the database, respectively, while the same measure for the Long Short-Term Memory model and Bidirectional Long Short-Term Memory model are 50%/38.9% (MAE/RMSE) and 61.1%/61.1% (MAE/RMSE), respectively. If we consider forecasting accuracy, computing expense, and interpretability, the Lee–Carter model with ARIMA exhibits the best overall performance, but the recurrent neural networks could also be good candidates for mortality forecasting for divisions in the United States.

**Keywords:** mortality prediction; Lee–Carter model; long short-term memory; bidirectional LSTM; gated recurrent unit

## 1. Literature Review

Modeling and forecasting future mortality rates are some of the most significant problems for life insurance, demography, and other social sciences. Many countries have experienced a rapid increase in life expectancy in recent decades (following the Second World War), and this increase has augmented the difficulties in modeling and predicting future mortality. Several stochastic mortality models have been proposed, for example, the famous Lee–Carter model (LC) by Lee and Carter [1]. Because of its simplicity, interpretability and, of course, convenience, the LC model has become the most frequently used stochastic model. Many improvements to the LC model have been proposed, including the Poisson extension of the LC model by Brouhns et al. [2], a functional data method using penalized regression by Hyndman and Ullah [3], a static PCA extension of the LC model by Shang [4], and a Two-Factor model for stochastic mortality with parameter uncertainty by Cairns et al. [5], also known as the Cairns–Blake–Dowd model (CBD model). Other studies related to the improvement of the Lee–Carter model can be found in the field of new techniques application, such as a cohort-based extension to the Lee–Carter model by Renshaw and Haberman [6] and the application of the random forest algorithm to improve the Lee–Carter mortality forecasting by Deprez et al. [7] and Levantesi and Pizzorusso [8].

Recently, with the development of machine learning, deep learning, and big data, new opportunities and challenges have been introduced into the actuarial field [9,10]. At

present, recurrent neural networks stand prominently on the stage of mortality forecasting. Recurrent neural networks are useful in recognizing unidentifiable patterns from a large dataset with many features. However, the application of neural networks in the mortality rate forecasting field have not offered as much insight as we expected. The biggest "problem" of neural networks is outcome uncertainty and the lack of demographic meaning. Moreover, instead of being explained by a specific hypothesis, neural networks are driven by data, which limits references to current works. However, many researchers still seek to apply neural networks for mortality forecasting. A neural network was proposed to predict and simulate the log-mortality rate by Hainaut [11]. This study showed that the neural network could catch more information from known mortality data and then duplicate the nonlinear trend in the prediction. Perla et al. [12] proposed a comparison study between the Lee–Carter model and deep learning models with data from Human Mortality Database (HMD) [13]. Some of the studies focus on applying deep learning models to predict the time index in the Lee–Carter model. Nigri et al. [14,15] applied the recurrent networks with Long Short-Term Memory (LSTM) architecture to predict the time index of the Lee–Carter model and found that neural networks forecasting could provide a high accuracy trend compared to the ARIMA models in several countries and for both genders. Marino and Levantesi [16] extended the neural networks approach by Nigri et al. [14] and derived the related confidence interval representing the Long Short-Term Memory model's parameter uncertainty. Richman and Wuthrich [17] compared the forecasting performance of a neural network extension of the Lee–Carter model and several different Lee–Carter approaches on all the countries in the HMD [13]. Other relevant applications of machine learning and deep learning studies in the mortality field can be found in Castellani et al. [18], Hong et al. [19], Richman and Wuthrich [20], and Gabrielli and Wuthrich [21].

## 2. Introduction

From the previous works, we noticed that most of the studies on neural networks applications examine Long Short-Term Memory. The Bidirectional Long Short-Term Memory and Gated Recurrent Unit could also be applied to time series forecasting tasks. Most of the studies propose recurrent neural network models as an approach to estimate the time index in LC model. This paper focuses, instead, on a direct comparison to the mortality rate prediction results between the Lee–Carter model and deep learning models. Gabor Petnehazi and Jozsef Gall [22] proposed a comparison study on mortality prediction with the LSTM model and the Lee–Carter model on countries all around world. We expand this comparison to three recurrent neural networks, LSTM, Bi-LSTM, GRU, and the Lee–Carter model. We also noticed that most of the studies explore mortality rates of the different countries around the world, so we chose to apply our experiments on the nine census divisions in the United States, which shows extraordinary prediction results of the LC model with ARIMA.

One problem we identified is in regard of the selection of parameters of the recurrent neural networks; some studies chosen the same parameters for different models to make a direct comparison on the forecasting abilities. However, in this study, we chose parameters with the best forecasting performance and compared their forecasting results.

Therefore, the key contribution of this paper is its novel comparison study of the nine census divisions' mortality rate predictions in the US using the Lee–Carter model, the Long Short-Term Memory model, the Bi-directional Long Short-Term Memory model, and the Gated Recurrent Unit. We measure the forecasting results by Mean Average Error (MAE) and Root-Mean-Square Error (RMSE) in an out-of-sample test.

The paper is organized in the following sections: The introduction of three different recurrent neural networks is presented in Section 3. The Lee–Carter model and Singular Value Decomposition (SVD) methods are presented in Section 4. Data features and data preprocessing information are shown in Section 5. Section 6 illustrates the numerical process of the experiments, and Section 7 offers the conclusion.

## 3. Recurrent Neural Networks

### 3.1. Long Short-Term Memory

The Long Short-Term Memory is an improved version of recurrent neural networks (RNN); it uses special units in addition to standard units. RNNs store information from the past by using the output of the previous unit as input. This means that it can cause problems when facing long-term data. As a result, Hochreiter and Schmidhuber [23] introduced a special RNN that could store long-term memory as well as discard useless memory. The special structure makes the Long Short-Term Memory model well suited to classification problems, regression problems, and especially on time-series tasks. In what follows, we investigated some mathematical functions in LSTM.

The sigmoid ($\sigma$) and the hyperbolic tangent (tanh) are the most frequently nonlinear activation functions in neural networks, as shown in Equations (1) and (2).

$$\text{Sigmoid}: \ \sigma(x) = \frac{1}{1 + e^{-x}} \tag{1}$$

$$\text{Tangent hyperbolic}: \ \mathbf{tanh}(x) = \frac{e^x - e^{-x}}{e^x + e^{-x}} \tag{2}$$

The sigmoid ($\sigma$) values range between 0 and 1, and it is used to decide if the information received should be uploaded and retained or be discarded and forgotten. Because any number that multiplies with 0 will be equal to 0, values may disappear or be considered as "forgotten". Any number that multiplies with 1 will remain the same, so it is "kept" or retained.

The activation function tangent hyperbolic (tanh) ranges between $-1$ and 1, and it is used to regulate the values flowing through the neural network.

A common LSTM unit has a 3-gate mechanism as will be discussed in the following section.

Forget gate

The forget gate controls the degree of information loss from the previous cell state; in another words, it decides what information is dropped or kept. The previous information passes through the sigmoid function, and the numbers coming out of the forget gate are between 0 and 1. Values closer to 0 are more likely to be forgotten, and values closer to 1 are more likely to be kept. We denoted the forget gate as $f_t$. Weight metrices for the forget gate are identified as $W_f$ *and* $U_f$, and the hidden units as $h_t$. The input variable $x = x_1, x_2, \ldots, x_T$ is a time series sequence at time $t = 1, 2, \ldots, T$, from which we could obtain Equation (3):

$$f_t = \sigma\left(W_f x_t + U_f h_{t-1}\right) \tag{3}$$

Input gate

The input gate controls the degree of new information to store in the current cell. It uses a sigmoid layer to decide what information would be updated in the current cell state and then uses a tanh layer to create new vector for the cell state. We denoted the input gate as $i_t$, the weight matrices for input gate as $W_i$ and $U_i$, and the hidden units as $h_t$. Thus, we have Equation (4):

$$i_t = \sigma(W_i x_t + U_i h_{t-1}) \tag{4}$$

Cell State

After the previous works, the old cell state is updated with the information collected from the gates. This step is achieved by multiplying the old cell state with the weight matrix generated by the forget gate and filter the original information to decide the kept and dropped parts. Then, the results are multiplied in the input gate to obtain the new information, which is added to the cell state. We used $C_t$ to represent the current cell state,

$\widetilde{C_t}$ represents the candidate cell state, $W_c$ and $U_c$ represent the weight matrices for cell statement, and the hidden units are $h_t$. The symbol $\odot$ is the Hadamard product. Here, we also provide a quick introduction of Hadamard product; for two matrices $A$ and $B$ with the same dimension, the Hadamard product is:

$$(A \odot B)_{ij} = A_{ij}B_{ij} \tag{5}$$

In the example of $3 \times 3$ matrix $A$ and $B$, we have:

$$\begin{bmatrix} a_{11} & a_{12} & a_{13} \\ a_{21} & a_{22} & a_{23} \\ a_{31} & a_{32} & a_{33} \end{bmatrix} \odot \begin{bmatrix} b_{11} & b_{12} & b_{13} \\ b_{21} & b_{22} & b_{23} \\ b_{31} & b_{32} & b_{33} \end{bmatrix} = \begin{bmatrix} a_{11}b_{11} & a_{12}b_{12} & a_{13}b_{13} \\ a_{21}b_{21} & a_{22}b_{22} & a_{23}b_{23} \\ a_{31}b_{31} & a_{32}b_{32} & a_{33}b_{33} \end{bmatrix} \tag{6}$$

Hence, we have Equations (7) and (8) for cell state:

$$\widetilde{C_t} = \tanh(U_c x_t + W_c h_{t-1}) \tag{7}$$

$$C_t = f_t \odot C_{t-1} + i_t \odot \widetilde{C_t} \tag{8}$$

Output gate

The output gate calculates the new output value of the current cell. The sigmoid layer is used again to generate the weight matrix and we used this matrix to decide what would be the output of the cell state. Then, the weight matrix is multiplied with the input results to output the part of the cell state. We denoted the output gate as $o_t$, and the weight matrices for output gate as $W_o$ and $U_o$, and we generated Equations (9) and (10):

$$o_t = \sigma(U_o x_t + W_o h_{t-1}) \tag{9}$$

$$h_t = o_t \odot \tanh(C_t) \tag{10}$$

The $U$s and the $W$s are the weight matrices that help in applying the model to different length of input data. A key point of the weight matrices is that they do not change over time. The same weight matrices are used in every time steps. The weight matrices $W_f$, $W_i$, $W_o$, $W_c$ have the same dimension (*input dimension $\times$ output dimension*) and the weight matrices $U_f$, $U_i$, $U_o$, $U_c$ have the same dimension (*output dimension $\times$ output dimension*). These matrices are learned using a variant of the gradient descent algorithm. Moreover, we can calculate the network parameters for a single layer with the following equation:

$$\begin{aligned} parameters = \quad & \mathbf{4} \times output\ dimension \\ & \times (output\ dimension + input\ dimension + \mathbf{1}) \end{aligned} \tag{11}$$

In the end, a single LSTM unit structure is shown in Figure 1.

### 3.2. Bi-directional Long Short-Term Memory

A new Long Short-Term Memory model, named Bi-directional Long Short-Term Memory (Bi-LSTM), was proposed by Schuster and Paliwa [24]. It is an extension of LSTM, with the main difference between Bi-LSTM and LSTM being that, instead of one forward direction hidden layer, the Bi-LSTM model uses two similar hidden layers with opposite directions. In the one forward direction, Bi-LSTM learns in increasing order of sequence input and, in the backward direction, it learns the information decreasing order of the sequence input. This means that both past and future information is utilized. However, compared to LSTM, the Bi-LSTM model requires more to finish the training, so it presents a considerable challenge to practice.

Bi-LSTM performs well in natural language-processing problems, such as sentence classification and translation. It could also be applied in handwritten recognition problem, sequence problems, and similar fields.

A Bi-LSTM unit is the same as the LSTM unit, but the architecture is different. To show the difference, the architectures of the LSTM and Bi-LSTM are shown in Figure 2. We can see that both past and future information from the dataset is used.

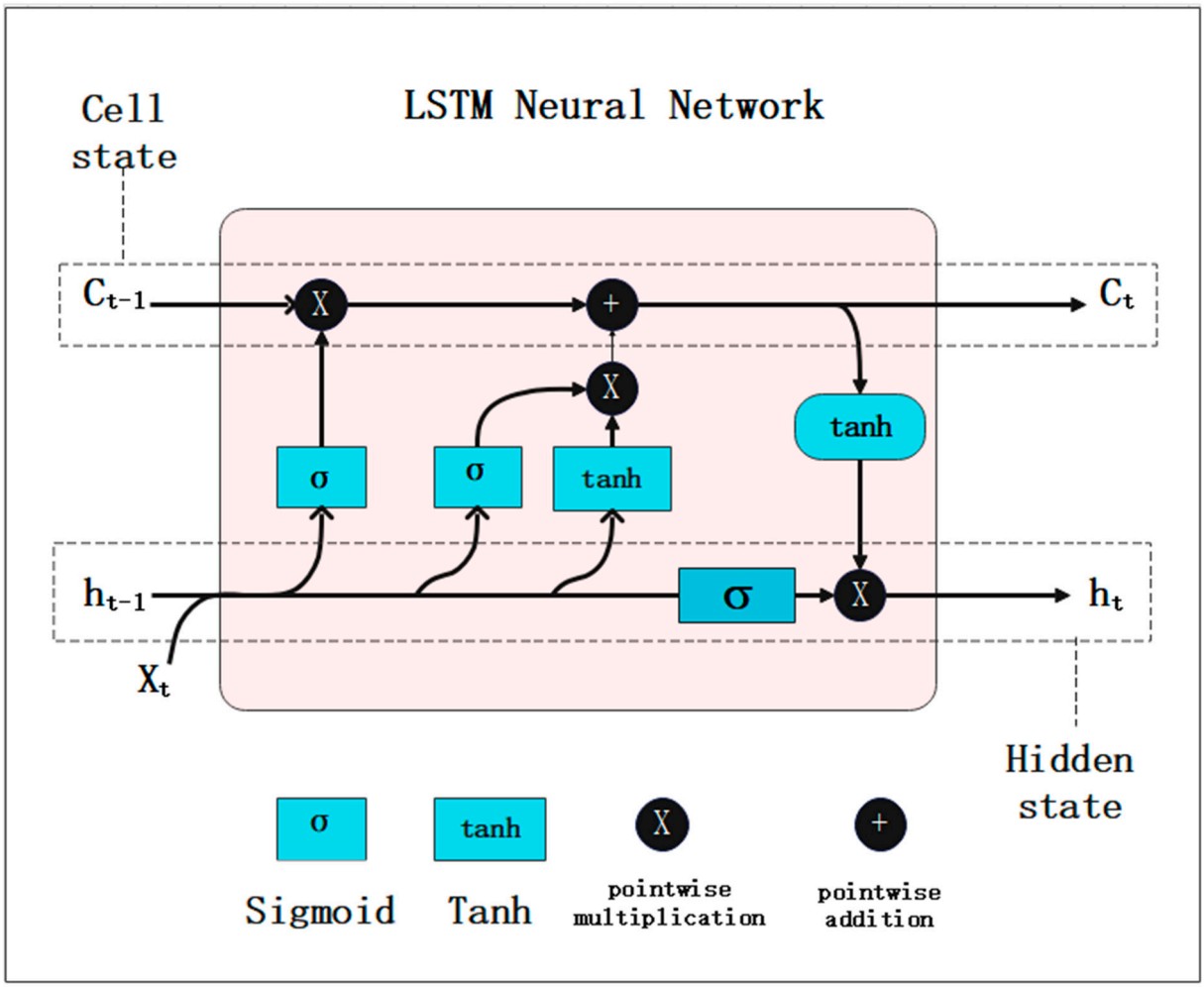

**Figure 1.** LSTM unit structure.

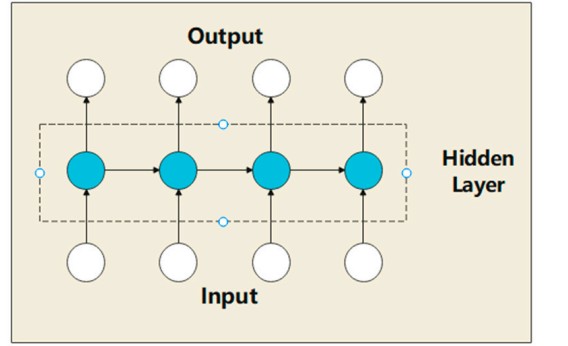 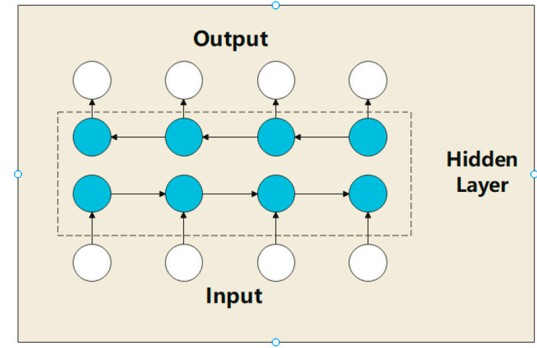

**Figure 2.** LSTM architecture vs. Bi-LSTM architecture.

### 3.3. Gated Recurrent Unit

The last type of recurrent neural network is the Gated Recurrent Unit (GRU), introduced by Kyunghyun Cho et al. [25]. It is similar to LSTM, but it has fewer parameters, gates, and equations. Generally speaking, it is really difficult to tell which network is the best model for the case. Other comparison studies between LSTM and GRU can be found

in Chung et al. [26]. The GRU model merges the forget gate and input gate of LSTM models into a single updated gate. A Gated Recurrent Unit works according to Equations (12)–(15).

$$z_t = \sigma(W_z x_t + U_z h_{t-1}) \tag{12}$$

$$r_t = \sigma(W_r x_t + U_r h_{t-1}) \tag{13}$$

$$\widetilde{h}_t = tanh(W_h x_t + U_h(r_t \odot h_{t-1})) \tag{14}$$

$$h_t = z_t \odot \widetilde{h}_t + (1 - z_t) \odot h_{t-1} \tag{15}$$

$z_t$ denotes the update gate and $r_t$ denotes the reset gate; $W$s and $U$s are the weight matrices; $h_t$ is the output information to the next unit; $\widetilde{h}_t$ is the current cell state; $x_t$ denotes the input vector; and the Hadamard product is $\odot$. The single gated recurrent unit structure is shown in Figure 3.

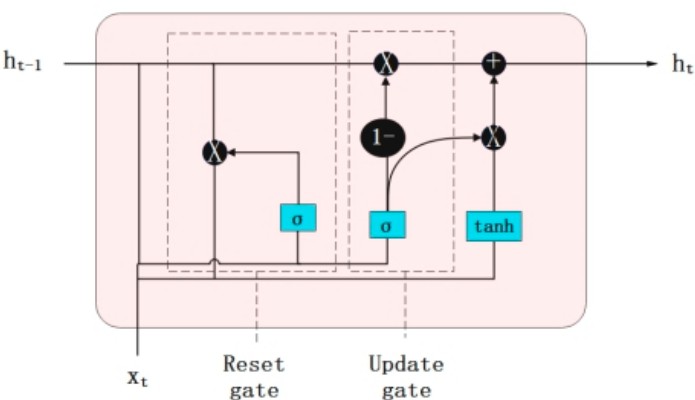

**Figure 3.** GRU unit structure.

## 4. Lee–Carter Model

In this section, we discuss some important concepts regarding the Lee–Carter model by [1]. The Lee–Carter model is a demographic model that is widely used in mortality prediction and life expectancy forecasting for different countries. The Lee–Carter model implies a linear relationship between the central mortality rate of an age group $m_{x,t}$ and age interval $x$ and year $t$. Equation (16) describes the model:

$$m_{x,t} = \exp(\alpha_x + \beta_x \kappa_t) \tag{16}$$

It can be rewritten as Equation (17):

$$\ln(m_{x,t}) = \alpha_x + \beta_x \kappa_t \tag{17}$$

where $\alpha_x$ represents the specific average log-mortality rate, $\beta_x$ means the deviation in mortality due to age profile $\kappa_t$ variations, and $\kappa_t$ is the time index to the year $t$. Another is that the Lee–Carter model is subject to the constraints on the parameters, so we have (18).

$$\sum_{x=x_1}^{x_p} \widehat{\beta_x} = 1 \ and \ \sum_{t=t_1}^{t_n} \widehat{\kappa_t} = 0 \tag{18}$$

In practice, Singular Value Decomposition (SVD), Maximum Likelihood estimation (MLE), and Least Square (LS) are the three classical methods to estimate the parameters of the Lee–Carter model. In this paper, we applied the Singular Value Decomposition (SVD) approach to the Lee–Carter model and use the ARIMA process to estimate the time index $k_t$.

The first step calculates the parameter $\alpha_x$, which is just the average values of raw $\ln(m_{x,t})$ (observation) over time, shown in Equation (19).

$$\widehat{\alpha_x} = \frac{1}{t} \sum_{t=t_1}^{t_n} \ln(m_{x,t}) \tag{19}$$

The estimation of $\beta_x$ and $\kappa_t$ is obtained by the singular value deposition of the matrix of $\ln(m_{x,t}) - \alpha_x$. Here, we present a quick introduction of the singular value decomposition: first, we denote a matrix $\ln(m_{x,t}) - \alpha_x = A$. Supposing that matrix $A$ is a $n \times m$ matrix, then $A$ could be computed as:

$$A_{n \times m} = U_{n \times n} D_{n \times p} V'_{p \times p} \tag{20}$$

where the $U$ and $V$ are orthogonal, $V'$ represents the $V$ transposed, and $D$ has the same dimensions as $A$ and has singular value. We calculated the $\hat{\beta}_x$ according to the following Equations (21) and (22):

$$\widehat{\beta_x} = \frac{U_{x,1}}{\sum U_{x,1}} \tag{21}$$

$$\widehat{\kappa_t} = V_{t,1} D_{1,1} \sum_{x=x_1}^{x_p} U_{x,1} \tag{22}$$

All the SVD process can be achieved in R with the function SVD. For the forecasting of the time index $\kappa_t$, we chose to use the traditional time series model ARIMA model with drift, which is discussed in an upcoming section.

## 5. Data

This study focused on mortality rates for the nine census divisions in the US: New England, Middle Atlantic, East North Central, West North Central, South Atlantic, East South Central, West South Central, Mountain, and Pacific. The data for the numerical experiment were collected from usa.mortality.org (the United States mortality database). This study was obtained on the life tables for nine census divisions in the US. This database provides the central mortality rates for 24 age groups (from 0 to 110+) by gender. These datasets were split into training and test sets with the rules of 80% training and 20% test. Due to the time series data, we could not randomly split the data set, so we picked the historical data as the training set and predicted the future mortality rates based on that. The total years and the corresponding testing set years of the data are shown in Table 1 and the average mortality rates by age groups and gender are shown in Table 2.

**Table 1.** Total and testing set years by regions.

| Census Division | Total Years | Testing Set Years |
|---|---|---|
| New England | 1966–2015 | 2006–2015 |
| Middle Atlantic | 1966–2015 | 2006–2015 |
| East North Central | 1966–2015 | 2006–2015 |
| West North Central | 1966–2015 | 2006–2015 |
| South Atlantic | 1966–2015 | 2006–2015 |
| East South Central | 1966–2015 | 2006–2015 |
| West South Central | 1966–2015 | 2006–2015 |
| Mountain | 1966–2015 | 2006–2015 |
| Pacific | 1966–2015 | 2006–2015 |

**Table 2.** Average mortality rates by age groups and gender.

| Age Group | Male | Female |
|:---:|:---:|:---:|
| 0 | 0.0103184 | 0.008126 |
| 1–4 | 0.0003978 | 0.0003192 |
| 5–9 | 0.000224 | 0.0001586 |
| 10–14 | 0.00024 | 0.0001582 |
| 15–19 | 0.0008736 | 0.0003356 |
| 20–24 | 0.0012676 | 0.0004144 |
| 25–29 | 0.0012794 | 0.0004938 |
| 30–34 | 0.0014682 | 0.0006756 |
| 35–39 | 0.0019484 | 0.0010024 |
| 40–44 | 0.0028846 | 0.0015864 |
| 45–49 | 0.004514 | 0.0025628 |
| 50–54 | 0.0071644 | 0.0039904 |
| 55–59 | 0.011396 | 0.0061842 |
| 60–64 | 0.0179772 | 0.0097272 |
| 65–69 | 0.0277216 | 0.0151632 |
| 70–74 | 0.0425512 | 0.0242644 |
| 75–79 | 0.065019 | 0.0394208 |
| 80–84 | 0.100258 | 0.065534 |
| 85–89 | 0.1565726 | 0.111995 |
| 90–94 | 0.2401684 | 0.1854848 |
| 95–99 | 0.3470992 | 0.289526 |
| 100–104 | 0.4723096 | 0.4209162 |
| 105–109 | 0.601285 | 0.5649978 |
| 110+ | 0.7013268 | 0.680411 |

## 6. Numerical Process

After predicting the parameters $\alpha_x$, $\beta_x$, and $\hat{\kappa}_t$ in the Lee–Carter model with the Singular Value Decomposition (SVD) method, we used the AutoRegressive Integrated Moving Average model (ARIMA) to predict the future $\hat{\kappa}_t$. The process of finding the best ARIMA model to univariate time series was achieved on the R vision 4.2.1 with the auto ARIMA in the forecast package. This technique is based on the Hyndman–Khandakar algorithm by Hyndman and Khandakar [27]. The idea is using a unit root test to test the stationarity of the time series and choose the degree of differencing d, and then select the best degree of auto-regressive p and moving average order q by the 2 criterion Akaike Information Criterion (AIC) and Bayesian Information Criterion (BIC). The best ARIMA (p,d,q) with drift for males and females are shown in the following Tables 3 and 4, respectively.

We tried to find out the parameters with the best performance in neural networks (at least in some ranges). However, we did not use any systematic process to search for the most reasonable parameters. In the end, a simple neural networks architecture was used, which contains a hidden layer and a dense layer with a single unit. In the model compilation part, we proposed the optimizer Adam and the loss function is mean square error (MSE). The recurrent neural networks predictions were run in Python with the TensorFlow package and the neurons, batch size, epochs, and the dropout percent for neural networks for females and males by divisions are shown in Tables 5 and 6, respectively.

**Table 3.** Best ARIMA(p,d,q) for females.

| Census Division | Best ARIMA(p,d,q) |
|---|---|
| New England | ARIMA(1,1,1) |
| Middle Atlantic | ARIMA(2,1,0) |
| East North Central | ARIMA(0,1,0) |
| West North Central | ARIMA(1,1,2) |
| South Atlantic | ARIMA(0,1,0) |
| East South Central | ARIMA(2,2,1) |
| West South Central | ARIMA(0,1,1) |
| Mountain | ARIMA(1,2,1) |
| Pacific | ARIMA(0,1,0) |

**Table 4.** Best ARIMA(p,d,q) for males.

| Census Division | Best ARIMA(p,d,q) |
|---|---|
| New England | ARIMA(0,1,0) |
| Middle Atlantic | ARIMA(0,1,0) |
| East North Central | ARIMA(0,1,0) |
| West North Central | ARIMA(1,1,0) |
| South Atlantic | ARIMA(0,1,0) |
| East South Central | ARIMA(0,1,0) |
| West South Central | ARIMA(0,1,0) |
| Mountain | ARIMA(0,1,0) |
| Pacific | ARIMA(0,1,0) |

To measure the prediction performance, we selected two error criteria for the out-of-sample test, mean absolute error (MAE), and root-mean-square error (RMSE). The equations of the MAE and RMSE are presented as Equations (23) and (24). The total amount of data in the test data are denoted by $n$, $\widehat{m_{x,t}}$ represents the predicted mortality rate, and $m_{x,t}$ is the actual mortality rate.

$$MAE = \frac{1}{n} \sum_{t=1}^{n} |\widehat{m_{x,t}} - m_{x,t}| \tag{23}$$

$$RMSE = \sqrt{\frac{1}{n} \sum_{t=1}^{n} (\widehat{m_{x,t}} - m_{x,t})^2} \tag{24}$$

Ten consecutive results were collected by each recurrent neural network and the average MAE and RMSE are shown in Table 7 by gender and divisions.

Considering the average MAE and RMSE by genders, every recurrent neural network approach offers a comparable performance to the LC model. Among them, the GRU model shows the best performance on both genders. The GRU model showed a better MAE value and RMSE value than the LC model at 72.2% (13/18) and 67.7% (12/18) of the database, respectively, and the LSTM and Bi-LSTM provided 50%/38.9% (MAE/RMSE) and 61.1%/61.1% (MAE/RMSE), respectively. It is surprising that LSTM did not have the good performance we expected before the experiment.

We also compared the averaged performance by genders between the models. A summary of the averaged MAE and RMSE values is shown in Table 8.

**Table 5.** The number of neurons, batch size, epochs, and dropout percent for the female database by divisions.

| Census Division | Neurons | Batch Size | Epochs | Dropout |
|---|---|---|---|---|
| New England | | | | |
| LSTM | 128 | 32 | 50 | 20% |
| Bi-LSTM | 128 | 32 | 50 | 20% |
| GRU | 128 | 32 | 50 | 20% |
| Middle Atlantic | | | | |
| LSTM | 64 | 32 | 100 | 30% |
| Bi-LSTM | 64 | 32 | 100 | 30% |
| GRU | 64 | 64 | 300 | 30% |
| East North Central | | | | |
| LSTM | 64 | 16 | 150 | 10% |
| Bi-LSTM | 64 | 16 | 150 | 10% |
| GRU | 64 | 16 | 150 | 10% |
| West North Central | | | | |
| LSTM | 128 | 32 | 50 | 20% |
| Bi-LSTM | 128 | 32 | 50 | 20% |
| GRU | 128 | 32 | 50 | 20% |
| South Atlantic | | | | |
| LSTM | 128 | 16 | 150 | 10% |
| Bi-LSTM | 128 | 16 | 150 | 10% |
| GRU | 128 | 16 | 300 | 20% |
| East South Central | | | | |
| LSTM | 128 | 32 | 300 | 10% |
| Bi-LSTM | 128 | 32 | 300 | 30% |
| GRU | 128 | 32 | 300 | 10% |
| West South Central | | | | |
| LSTM | 128 | 64 | 300 | 10% |
| Bi-LSTM | 64 | 64 | 300 | 10% |
| GRU | 128 | 16 | 300 | 10% |
| Mountain | | | | |
| LSTM | 128 | 32 | 50 | 20% |
| Bi-LSTM | 128 | 32 | 50 | 20% |
| GRU | 128 | 32 | 50 | 20% |
| Pacific | | | | |
| LSTM | 128 | 32 | 100 | 20% |
| Bi-LSTM | 128 | 32 | 100 | 20% |
| GRU | 128 | 32 | 100 | 20% |

We can see that the deep learning models have better performance on the female dataset. Simultaneously, the MAE and RMSE analysis showed that the LSTM and Bi-LSTM models are not effective on the male case prediction. Considering the average MAE and RMSE measurements, GRU offered the best prediction performance with 0.003946/0.008871 (MAE/RMSE). Examples of life expectancy predicted values are shown in Figures 4–7, which considered both genders; the Mountain division confirmed this gender difference in the prediction performance. The first 40 years were the training set and the last 10 years were the test set. Here, we picked the age groups of 40–44 and 90–94, which, respectively, represent the middle age and elderly groups.

The results show that the deep learning models are capable of displaying more details of the dataset with a nonlinear trend. The LC model sometimes underrates or overrates the future mortality rate (in most cases, it underpredicts; see Bergeron-Boucher et al. [28] and Booth et al. [29]). When we consider data with rapid changes (see the example of the 90–94 age group/male), the mortality rates remained stable for a period (year 20–year 40),

and no model successfully predicted the sudden decrease in the coming 10 years. The uncertainty of the future is still a challenge for the time series tasks.

**Table 6.** The number of neurons, batch size, epochs, and dropout percent for the male database by divisions.

| Census Division | Neurons | Batch Size | Epochs | Dropout |
|---|---|---|---|---|
| New England | | | | |
| LSTM | 32 | 16 | 300 | 10% |
| Bi-LSTM | 32 | 16 | 300 | 10% |
| GRU | 128 | 32 | 150 | 20% |
| Middle Atlantic | | | | |
| LSTM | 128 | 32 | 60 | 30% |
| Bi-LSTM | 64 | 32 | 60 | 30% |
| GRU | 64 | 16 | 100 | 30% |
| East North Central | | | | |
| LSTM | 128 | 64 | 50 | 10% |
| Bi-LSTM | 128 | 64 | 50 | 10% |
| GRU | 128 | 64 | 50 | 10% |
| West North Central | | | | |
| LSTM | 64 | 16 | 150 | 30% |
| Bi-LSTM | 128 | 32 | 150 | 30% |
| GRU | 64 | 32 | 150 | 30% |
| South Atlantic | | | | |
| LSTM | 64 | 32 | 300 | 10% |
| Bi-LSTM | 64 | 32 | 300 | 10% |
| GRU | 64 | 32 | 300 | 10% |
| East South Central | | | | |
| LSTM | 128 | 16 | 50 | 30% |
| Bi-LSTM | 128 | 32 | 50 | 30% |
| GRU | 64 | 32 | 300 | 30% |
| West South Central | | | | |
| LSTM | 64 | 16 | 30 | 10% |
| Bi-LSTM | 32 | 16 | 30 | 10% |
| GRU | 64 | 16 | 100 | 30% |
| Mountain | | | | |
| LSTM | 128 | 32 | 100 | 20% |
| Bi-LSTM | 128 | 32 | 100 | 20% |
| GRU | 128 | 32 | 100 | 20% |
| Pacific | | | | |
| LSTM | 32 | 32 | 300 | 30% |
| Bi-LSTM | 32 | 16 | 50 | 30% |
| GRU | 64 | 32 | 300 | 30% |

We also considered the prediction in a single year; this is shown with the example of the New England division case. We chose the year 2015 with the predictions of mortality rate and log-mortality rate, as shown in Figures 8–13, by genders. Figures 8 and 9 compare all the models in terms of mortality rate and log-mortality rate in one plot, and the remaining figures compare the log-mortality (*y*-axis) of LC, LSTM, Bi-LSTM, and GRU to the real data for all the age groups (*x*-axis), respectively.

**Table 7.** MAE and RMSE for the LC, LSTM, Bi-LSTM, and GRU by gender and divisions.

| Census Division | Female | | Male | |
|---|---|---|---|---|
| New England | MAE | RMSE | MAE | RMSE |
| LC | 0.003580 | 0.0085774 | 0.0038145 | 0.007061 |
| LSTM | 0.003333 | 0.0077581 | 0.003602 | 0.007446 |
| Bi-LSTM | 0.003559 | 0.0084523 | 0.004280 | 0.008178 |
| GRU | 0.003222 | 0.007591 | 0.004250 | 0.009505 |
| Middle Atlantic | MAE | RMSE | MAE | RMSE |
| LC | 0.002296 | 0.0055494 | 0.003419 | 0.0064182 |
| LSTM | 0.005479 | 0.012104 | 0.0036423 | 0.0070882 |
| Bi-LSTM | 0.004609 | 0.0107834 | 0.0045392 | 0.0093438 |
| GRU | 0.004957 | 0.0115375 | 0.0024576 | 0.0048186 |
| East North Central | MAE | RMSE | MAE | RMSE |
| LC | 0.004458 | 0.0106013 | 0.0042796 | 0.0081338 |
| LSTM | 0.002742 | 0.0054024 | 0.0050855 | 0.0117587 |
| Bi-LSTM | 0.002667 | 0.0056034 | 0.0056478 | 0.0103892 |
| GRU | 0.003531 | 0.0080238 | 0.0045146 | 0.0104677 |
| West North Central | MAE | RMSE | MAE | RMSE |
| LC | 0.006313 | 0.0147076 | 0.0058709 | 0.0123197 |
| LSTM | 0.004541 | 0.0104502 | 0.0050320 | 0.0095187 |
| Bi-LSTM | 0.004225 | 0.0100399 | 0.0038613 | 0.0073576 |
| GRU | 0.004378 | 0.0104372 | 0.0029962 | 0.0055895 |
| South Atlantic | MAE | RMSE | MAE | RMSE |
| LC | 0.004249 | 0.0100673 | 0.0043421 | 0.007902 |
| LSTM | 0.004162 | 0.0096163 | 0.0065645 | 0.0129754 |
| Bi-LSTM | 0.003537 | 0.0079331 | 0.0041443 | 0.0077775 |
| GRU | 0.004525 | 0.0103644 | 0.0042279 | 0.0087472 |
| East South Central | MAE | RMSE | MAE | RMSE |
| LC | 0.005919 | 0.0137948 | 0.006056 | 0.0121139 |
| LSTM | 0.006389 | 0.0154277 | 0.0062494 | 0.0135819 |
| Bi-LSTM | 0.006630 | 0.0161339 | 0.0043764 | 0.0091593 |
| GRU | 0.006237 | 0.0150568 | 0.003344 | 0.0074549 |
| West South Central | MAE | RMSE | MAE | RMSE |
| LC | 0.003881 | 0.0094994 | 0.004401 | 0.008112 |
| LSTM | 0.002977 | 0.0067081 | 0.008326 | 0.0186121 |
| Bi-LSTM | 0.002770 | 0.0061035 | 0.0088042 | 0.0187601 |
| GRU | 0.003814 | 0.0089701 | 0.0031701 | 0.0062397 |
| Mountain | MAE | RMSE | MAE | RMSE |
| LC | 0.005875 | 0.0136075 | 0.0058631 | 0.0116347 |
| LSTM | 0.005474 | 0.0129507 | 0.0055829 | 0.0130083 |
| Bi-LSTM | 0.005256 | 0.0124847 | 0.0037339 | 0.0076257 |
| GRU | 0.005158 | 0.0123561 | 0.0048700 | 0.0112312 |
| Pacific | MAE | RMSE | MAE | RMSE |
| LC | 0.00303 | 0.0063291 | 0.0038562 | 0.0073431 |
| LSTM | 0.00337 | 0.0069403 | 0.0045788 | 0.0090681 |
| Bi-LSTM | 0.002647 | 0.0054352 | 0.0055415 | 0.0105694 |
| GRU | 0.002453 | 0.0056640 | 0.0029244 | 0.0056143 |

**Table 8.** The averaged MAE and RMSE values for the models by genders.

| Model | MAE Female | MAE Male | RMSE Female | RMSE Male | Averaged MAE | Averaged RMSE |
|---|---|---|---|---|---|---|
| LC | 0.0044 | 0.04656 | 0.010304 | 0.009004 | 0.004528 | 0.009654 |
| LSTM | 0.004274 | 0.005407 | 0.009706 | 0.011451 | 0.004841 | 0.010579 |
| Bi-LSTM | 0.003989 | 0.004992 | 0.009219 | 0.009907 | 0.00449 | 0.009563 |
| GRU | 0.004253 | 0.003639 | 0.010000 | 0.007741 | 0.003946 | 0.008871 |

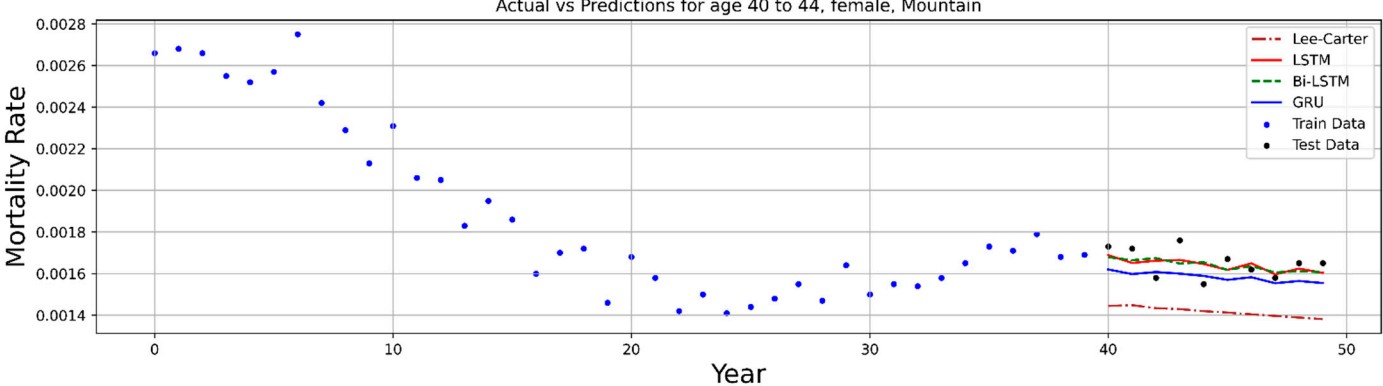

**Figure 4.** The life expectancy of models in the age group 40–44, Mountain female population.

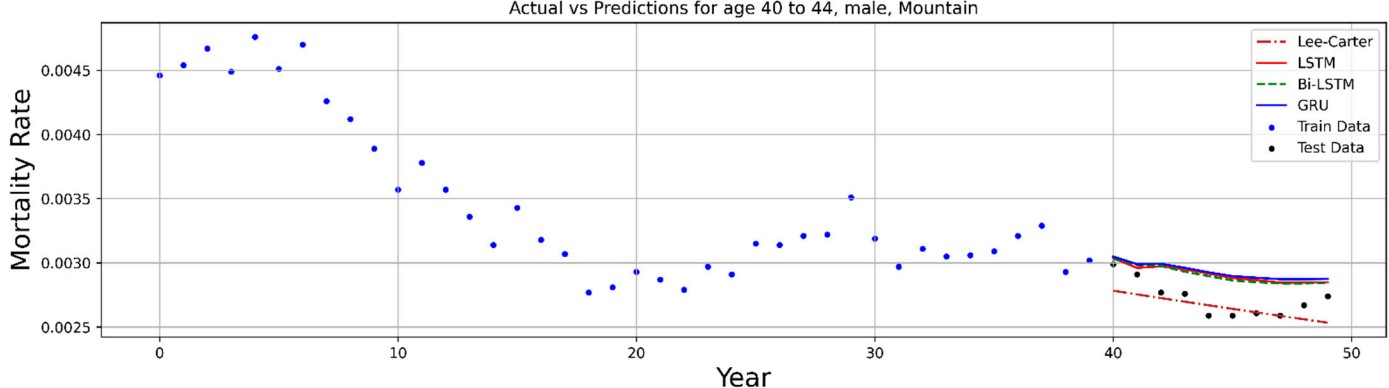

**Figure 5.** The life expectancy of models in the age group 40–44, Mountain male population.

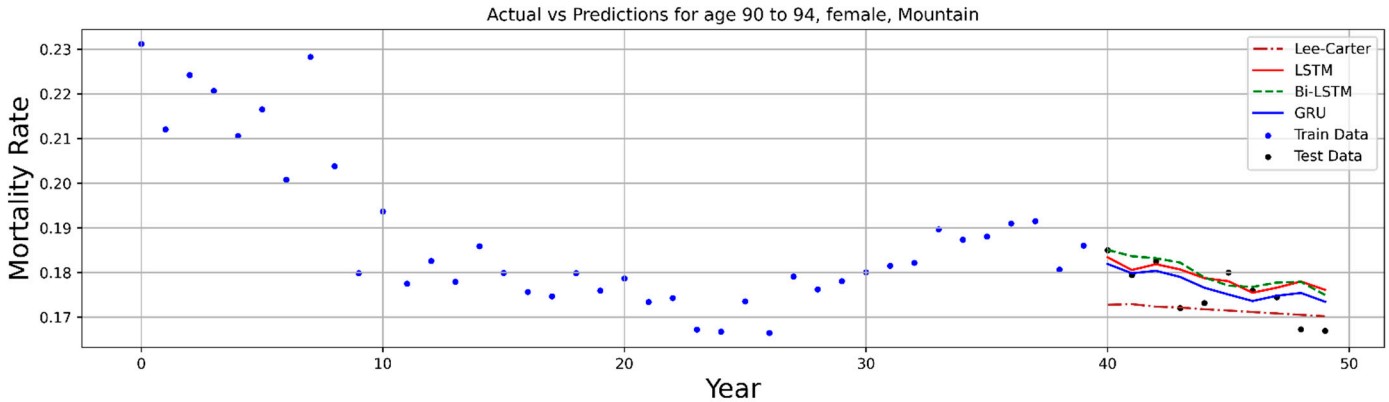

**Figure 6.** The life expectancy of models in the age group 90–94, Mountain female population.

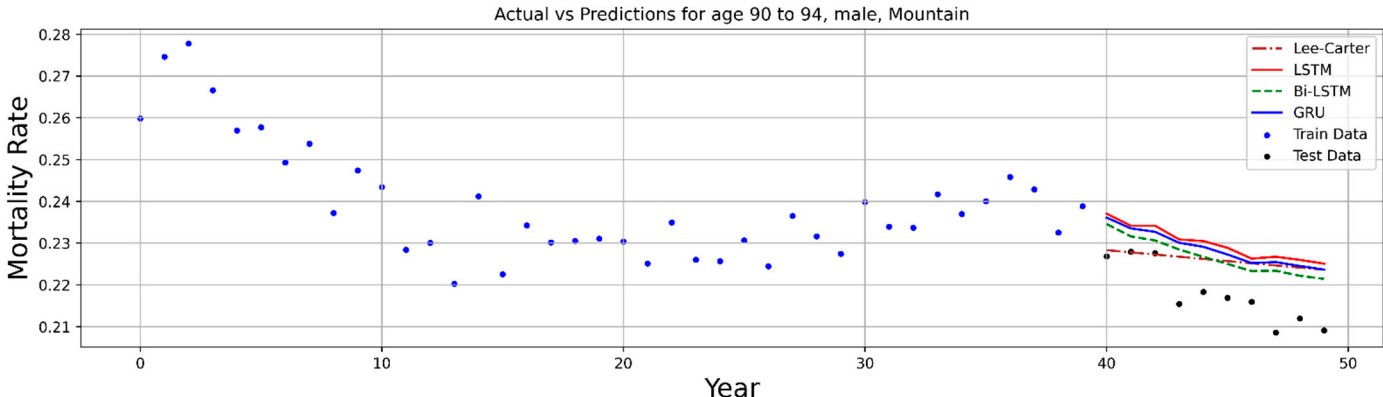

**Figure 7.** The life expectancy of models in the age group 90–94, Mountain male population.

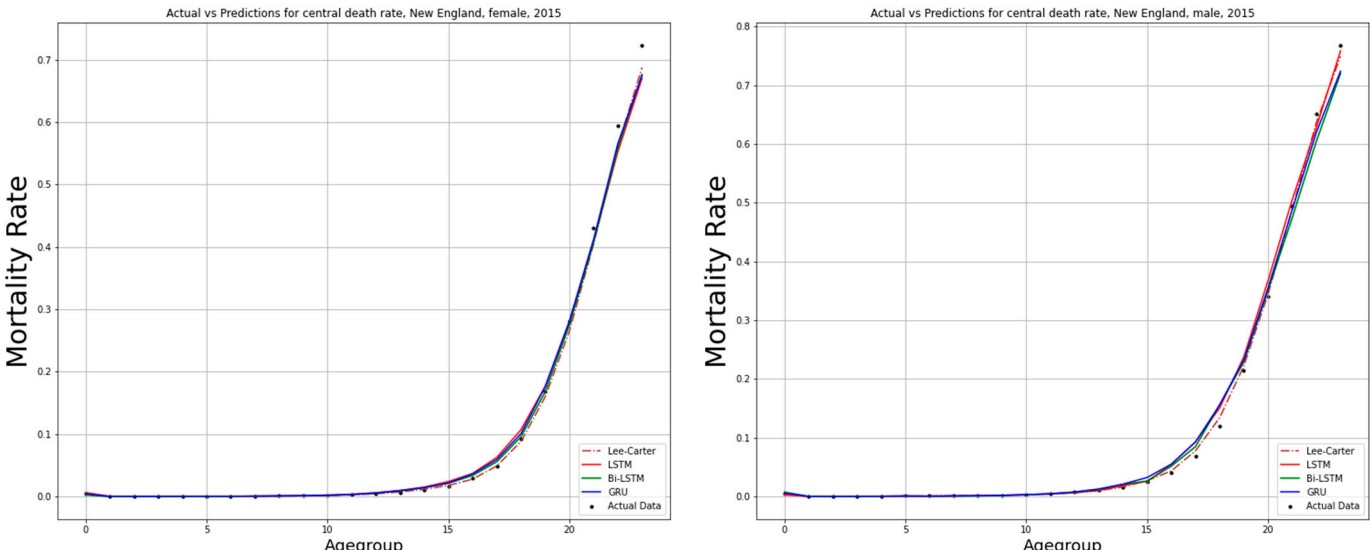

**Figure 8.** The predictions of the mortality rate for New England female (**left**) and male (**right**), 2015.

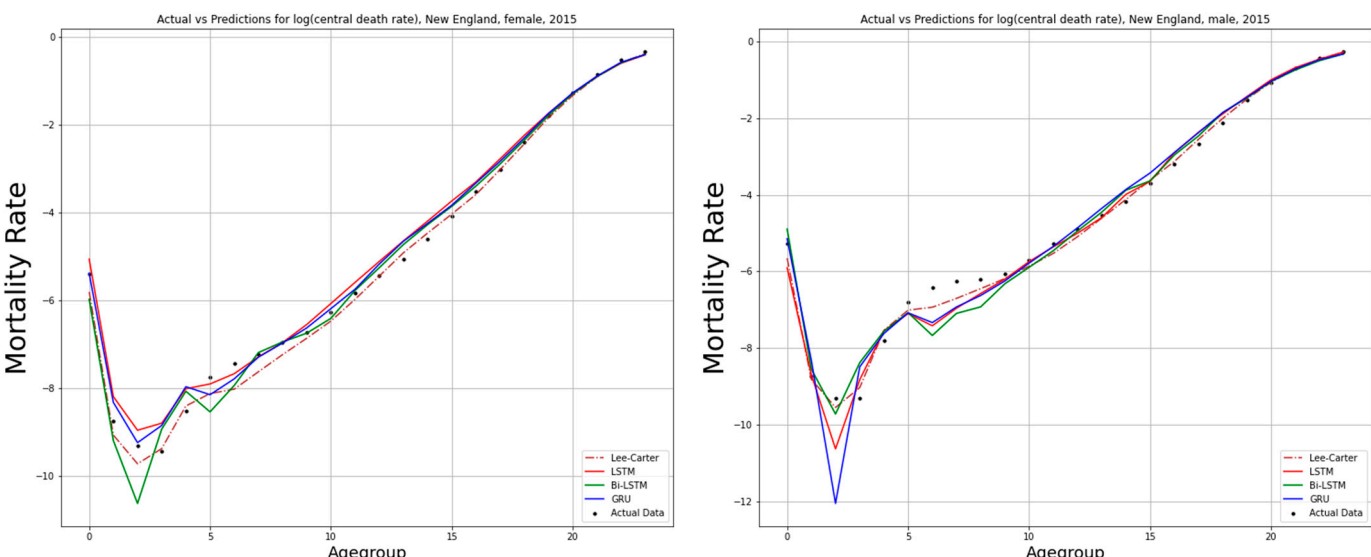

**Figure 9.** The predictions of the log-mortality rate for New England female (**left**) and male (**right**), 2015.

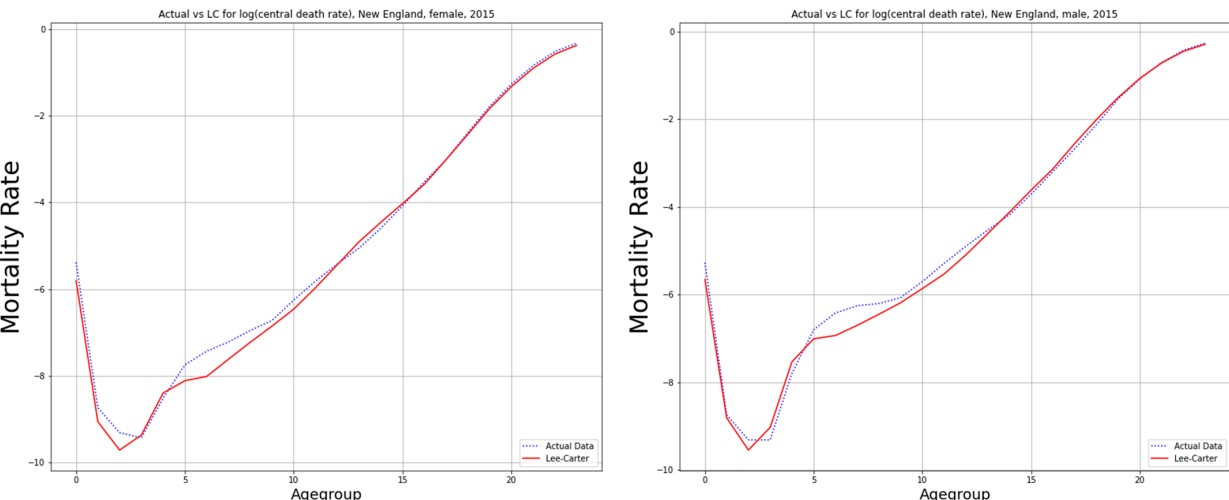

**Figure 10.** The predictions of the log-mortality rate for New England female (**left**) and male (**right**), 2015: LC (red) vs. actual (blue).

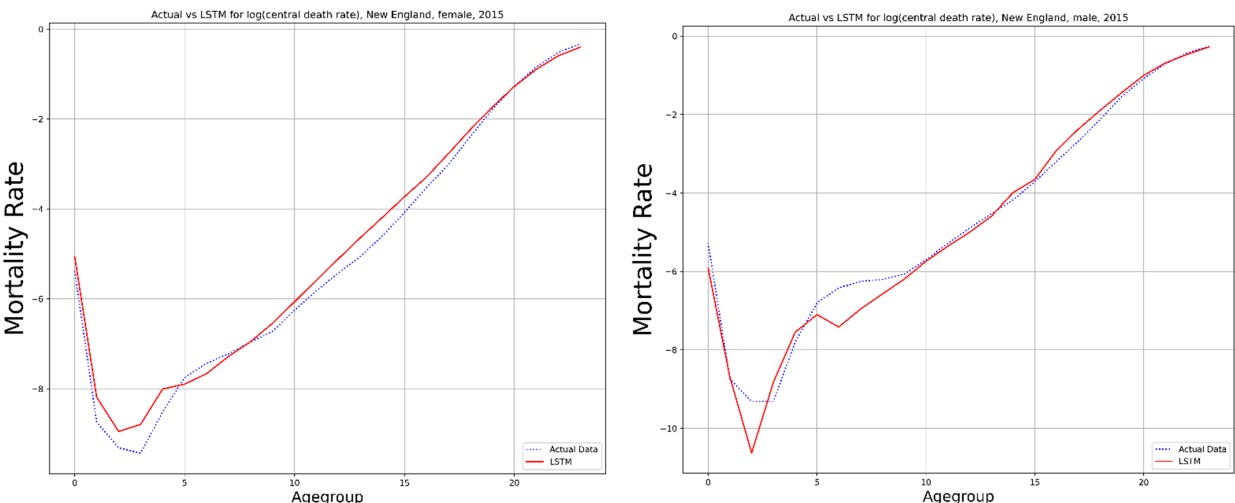

**Figure 11.** The predictions of the log-mortality rate for New England female (**left**) and male (**right**), 2015: LSTM (red) vs. real (blue).

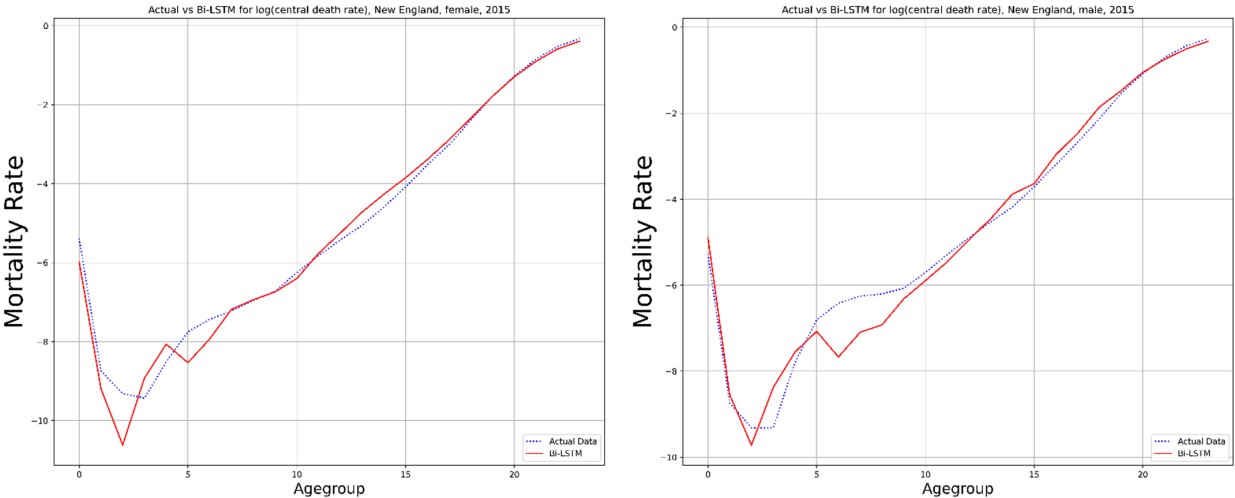

**Figure 12.** The predictions of the log-mortality rate for New England female (**left**) and male (**right**), 2015: Bi-LSTM (red) vs. real (blue).

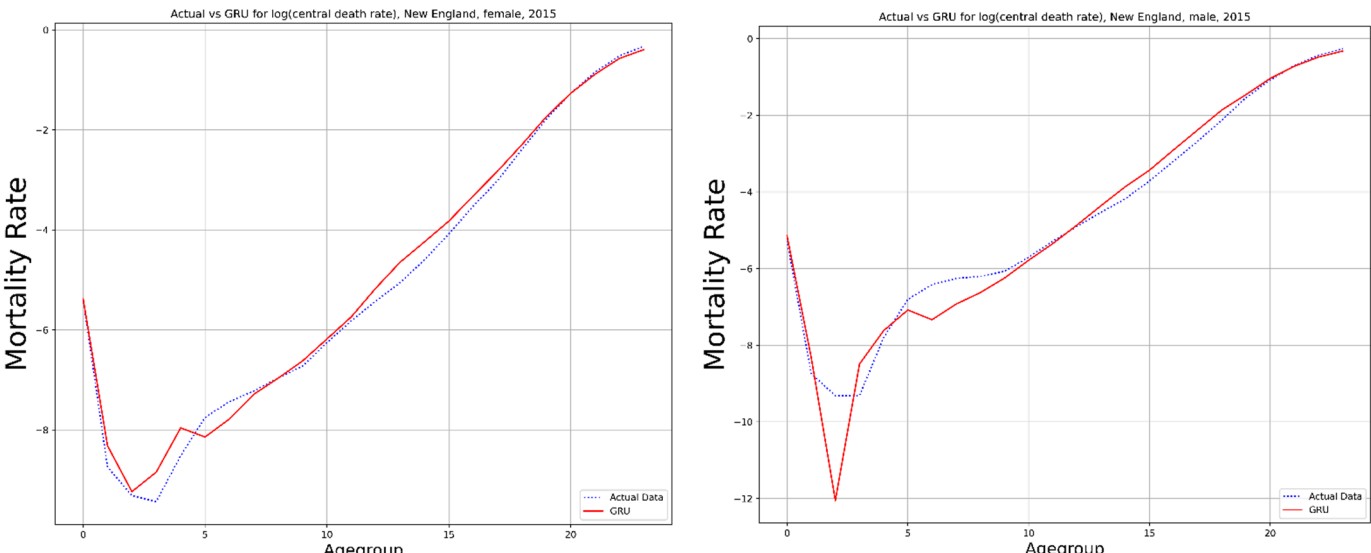

**Figure 13.** The predictions of the log-mortality rate for New England female (**left**) and male (**right**), 2015: GRU (red) vs. real (blue).

We noticed an excessively parabolic trend at the bottom of the log-mortality rate bathtub curve with the deep learning models, especially on male population, such as LSTM (male) and GRU (male).

## 7. Conclusions and Discussion

In the present paper, we proposed three popular RNN models, the LSTM model, Bi-LSTM model, and GRU model, for forecasting mortality rates. The experiment was performed on nine census divisions of the US according to gender. The results of the proposed comparative study show that GRU model obtained the best overall prediction values in the models.

By examining the comparison between the neural networks and the LC model, could we say that the neural networks perform better than the LC model?

On the one hand, the Bi-LSTM and GRU models had a better performance in terms of MAE and RMSE. On trend prediction, compared to the linear model, better prediction curves were displayed by the neural networks. Due to their unique architectures, more details could be caught, memorized, and replicated in the trend prediction.

On the other hand, even the GRU model could not show a high accuracy level in the mortality rate trend prediction. In other words, the deep learning models do not significantly improve the accuracy of mortality rate prediction when compared to the LC model. Regarding the algorithm itself, neural networks do not have the simplicity and interpretability of the LC model. The deep learning models are driven by data, and their random outcomes lack demographic meaning.

Moreover, according to the experiment, we noticed that the neural networks have better prediction performance on the female population than on the male population in the United States.

This experiment could serve as a reference for other works with the following as some potential improvements worthy of consideration. Firstly, some existing studies showed that the LC model has a worse performance for long-term mortality rate prediction than the short-term prediction results. This is a problem of the fitting period selection; many studies prefer to choose a shorter period to avoid data volatility. For example, Hyndman and Booth [30] used 1950 as the starting year to avoid the difficulties of the war years and the 1918 Spanish influenza pandemic. Other related studies are by Tuljapurkar et al. [31] and Lee and Miller [32]. The selected time period in our study was 1966–2015 or 50 years. The annual mortality rate training set was not be considered long term demographically,

especially avoiding the excess mortality rates by two world wars and the COVID-19 pandemic. That is one of the reasons that the LC model showed an incredible forecasting. Second, some of the existing research implied that achieving the prediction of log-mortality rates might demonstrate a better performance than on the mortality rate itself (the objective of the LC model). Third, according to the structure of the LC model, parameters $\alpha_x$ and $\beta_x$ are determined by data; they are the constant coefficients in the LC model, so the comparison of the prediction resembles more an indirect comparison of the best ARIMA models and neural networks.

Specifically, the uncertainty in the recurrent neural networks could be considered the most considerable challenge in the applications. The recurrent neural networks can provide predictions without any indication of variability. Some researchers aim to solve this problem through the construction of a confidence interval, such as Keren et al. [33], who proposed empirical calibration and temperature scaling for acquiring calibrated predictions intervals for neural network regressors. Khosravi et al. [34] wrote a comprehensive review for the prediction intervals. Several techniques are mentioned, such as bootstraps and Bayesian methods, but they have high computing expenses; these studies can be found in the works of Efron and Tibshirani [35], Dietterich [36], Heskes [37], and Gábor Petneházi [38]. According to these studies, there does not exist a reliable method to handle this problem on time series tasks to date. However, we can consider the results of the recurrent neural networks as good candidates for predicting mortality trends in the future.

Regarding future works, we believe that mortality rate trend prediction can be improved by combining other stochastic mortality models with more deep learning models or by testing the neural networks on different data. One popular study replaces the ARIMA model with the deep learning models and builds a LC-RNN model. As we mentioned, most of the studies focus on LSTM, but other deep learning models should be applied to the field of mortality prediction.

**Author Contributions:** Conceptualization, Y.C. and A.Q.M.K.; methodology, A.Q.M.K.; software, Y.C.; validation, A.Q.M.K. and Y.C.; formal analysis, Y.C.; investigation, Y.C.; data curation, A.Q.M.K.; writing—original draft preparation, Y.C.; writing—review and editing, A.Q.M.K.; visualization, Y.C.; supervision, A.Q.M.K. All authors have read and agreed to the published version of the manuscript.

**Funding:** This research received no external funding.

**Institutional Review Board Statement:** Not applicable.

**Informed Consent Statement:** Not applicable.

**Data Availability Statement:** Human Mortality Database. University of California, Berkeley (USA) and Max Planck Institute for Demographic Research (Germany). Available online: http://www.mortality.org (accessed on 15 September 2022).

**Conflicts of Interest:** The authors declare no conflict of interest.

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
