# Peer review of "Comparative Study of Mortality Rate Prediction Using Data-Driven Recurrent Neural Networks and the Lee–Carter Model"

_2504-2289, doi:10.3390/bdcc6040134_

Round 1

Reviewer 1 Report

Please follow the suggestions.

1. Please include more seminal work to conduct a fair literature review. you can use some part of your introduction section in it.

2. cite your identified research gaps after performing a literature review. Also, how did you try to bridge that gap? two or more sentences are enough to answer this question.

3. Cite clearly the key contributions of your paper.

4. in the present form the paper seems like a project report rather than a research paper hence, please conduct a literature review on a fairly amount of relevant papers and introduce that section in this paper.

Thank you for submitting the paper.

Author Response

First of all, thank you for all the comments. We have uploaded a revision manuscript, please have a look. We have enriched the literature review, and also cited some related studies as example in the introduction and conclusion part. About the research gap and the key contributions, We wrote a paragraph to explain most of the current studies  are focus on the LSTM model and country mortality rates, so we try to apply more recurrent neural networks to the divisions' mortality rates. Is this what you suggesting? Please let us know if that's further comments or we didn't make a good response to the previous comments.

Author Response

First of all, thank you for all the comments, I have uploaded a revision manuscript, please have a look. Your comments are really helpful.

Now I would reply to the comments one by one.

  1. Thank you for pointing out the logically disconnected, I tried my best to check the paper and rewrote some part, if there's still some logical problems, please point out.
  2. I also cited more research works in the paper.
  3. I added more explanations on the input and the dimension of the weight matrices, and yes it is the Hadamard product, I also put a quick introduction of it into the introduction part.
  4. The error component in the equation 14 has been deleted, you are right,  the estimated parameters have been added the symbol hat.
  5. I added the introduction of the SVD method, which will explain the U and V
  6. In the performance evaluation part, I have changed the equations into the equations with m, not h.
  7. I added age group 90-94 into the figures
  8. A paragraph about the uncertainty and  confidence interval has been added to the conclusion part.
  9. All the references have been checked and be written in the template way.

That's pretty much my response to the comments, please read the new manuscript and give me more feedback, thank you.

Author Response

First of all, thank you for the comments, I have uploaded the new manuscript, please read it, I would give some response right here.

I added a paragraph to explain the weighted matrices in the neural networks part, they are helping to fit the model to different length of input and output, the way to calculate the the dimension has been showed in the paper.

The titles of the tables and the figures went above.

About the LSTM does not have good performance as expect, the reason goes into my mind is that, that's potentially better parameters for LSTM model (the system method to find the "best " parameters would be computing expensive and time consuming, so we didn't do that), also, it's not big difference between the LSTM model and other models, it could not be considered as a bad model in this case. The LC model was invented based on the US mortality rates, it's not surprise that the LC model has a good forecasting performance. The reason we expected it could well fitting in this case just because it could handle some similar cases, again it doesn't have bad performance in this case study, just slightly worse than the other models.

About the Lee Carter model shows lower than the other predictions, it calculated by forecasting the time index, and then uses the time indexes the other two parameters from the historical data to calculate the future mortality rates. So it might be over predict or under predict the future mortality rates. By the way, yes it's linear. I added more details there, including the similar results from the other studies and another age group 90-94, from the 90-94 age group you can, in that case the LC model over predict the mortality rates.

 Thank you for your suggestions, I changed the actual data into dots in the figures and the blue line stands for the actual data, I added it into the foot of the figures.

Thank you again for your comments.

Reviewer 4 Report

The authors developed a comparison study about three mortality prediction models, including LSTM, LSTM with GRU, BiLSTM, and LC model. The models are well presented and explained in the manuscript. However, the manuscript needs to be improved in several aspects.

1. The comparison results should be discussed in the abstract.

2. Authors need to clearly present how samples were assigned to training and testing cohorts.  Line 211: These data will be split into training set and test set with the rules of 80% training and 20% test.  However, Table1 shows the Testing set years. Were data split by percentage or by years?

3. Line 193: use the ARIMA process to estimate the time index Kt. Why ARIMA model was used to estimate Kt paramenter.

4. Line 330:  some existing studies showed that the LC model has a worse performance on the long-term mortality rate prediction than the short-term prediction results.  Authors need to give references.

5 Figure 4-10, the X-axis needs to be explained clearly.

Author Response

First of all, thank you for all the comments. I have uploaded a new manuscript, please read. I would respond to the comments as following:

I have adjusted my abstract and  added some discussion in there, I'm not very sure is that good enough, please give me advance comments.

I explained the sets in the new manuscript, we can't randomly split the dataset in a timeseries data, so we could only split it on the time, by percentage, the first 40 years (80%)  as training set and the last 10 years (20%) as test set.

About the ARIMA, first it's a model for time series prediction, so we used ARIMA to predict the kappa in this case, another reason and the main reason we selected the ARIMA model is that the traditional and the most widely used method for LC model is ARIMA, we want to show a comparison between the new method and the old method.

I cited the related works at that part.

I added more descriptions to the figures in words.

Please read this new manuscript, thanks again for your comments.

Round 2

Reviewer 1 Report

The expectation was to frame the literature review section separately from the introduction section. However, the authors did include some relevant papers.

Author Response

Thanks for the comments. I restructured that part, now we show the literature review first and then the introduction, but that's not big changes in the contain. Is that good enough? If not, please point out. Thanks.

Reviewer 2 Report

The manuscript was edited in accordance with the recommendations provided by the referee in the previous round. In the present form, the manuscript is well written, has an adequate level of methodological formalization of the problem, and the empirical application is well described, both in its design and in its results. In conclusion, I believe that the manuscript is eligible for publication. I have no other recommendations for the authors.

Author Response

Thanks for the comments. I would still upload the revised vision of manuscript,  thank you.

Reviewer 4 Report

The accuracy performance should be reported in the abstract. So that audience can have the first impression of the author's work and contribution.

Author Response

Thanks again for the comments, I added the comparison results (the percentage numbers) of the neural networks and the Lc model in the abstract, is that good enough? If not, please point out again. Thanks
